# *BocODD1* and *BocODD2* Regulate the Biosynthesis of Progoitrin Glucosinolate in Chinese Kale

**DOI:** 10.3390/ijms232314781

**Published:** 2022-11-26

**Authors:** Shuanghua Wu, Ting Zhang, Yudan Wang, Muxi Chen, Jianguo Yang, Fei Li, Ying Deng, Zhangsheng Zhu, Jianjun Lei, Guoju Chen, Bihao Cao, Changming Chen

**Affiliations:** 1Key Laboratory of Biology and Germplasm Enhancement of Horticultural Crops in South China, Ministry of Agriculture and Rural Areas, College of Horticulture, South China Agricultural University, Guangzhou 510642, China; 2Hunan Vegetable Research Institute, Hunan Academy of Agricultural Science, Changsha 410125, China; 3Institute of Horticulture, Guizhou Academy of Agricultural Sciences, Guiyang 550000, China

**Keywords:** progoitrin glucosinolate, ODD, biosynthesis of glucosinolates, gene function, Chinese kale

## Abstract

Progoitrin (2-hydroxy-3-butenyl glucosinolate, PRO) is the main source of bitterness of *Brassica* plants. Research on the biosynthesis of PRO glucosinolate can aid the understanding of the nutritional value in *Brassica* plants. In this study, four ODD genes likely involved in PRO biosynthesis were cloned from Chinese kale. These four genes, designated as *BocODD1–4*, shared 75–82% similarities with the *ODD* sequence of *Arabidopsis*. The sequences of these four *BocODD*s were analyzed, and *BocODD1* and *BocODD2* were chosen for further study. The gene *BocODD1,2* showed the highest expression levels in the roots, followed by the leaves, flowers, and stems, which is in accordance with the trend of the PRO content in the same tissues. Both the expression levels of *BocODD1,2* and the content of PRO were significantly induced by high- and low-temperature treatments. The function of *BocODDs* involved in PRO biosynthesis was identified. Compared with the wild type, the content of PRO was increased twofold in the over-expressing *BocODD1* or *BocODD2* plants. Meanwhile, the content of PRO was decreased in the *BocODD1* or *BocODD2* RNAi lines more than twofold compared to the wildtype plants. These results suggested that *BocODD1* and *BocODD2* may play important roles in the biosynthesis of PRO glucosinolate in Chinese kale.

## 1. Introduction

Glucosinolates (GSLs) are a group of important secondary metabolites that are mainly found in the Brassicaceae family [1,2]. These secondary metabolites can protect plants from insects and some pathogen infections [3,4]. The degradation products of some GSLs, such as glucoraphanin (RAA), have strong anticarcinogenic activity [5,6]. However, another glucosinolate, PRO, is the main source of bitterness of *Brassica* plants, and the hydrolyzed production of PRO may cause goiters in mammals [7]. For these reasons, the accumulation of PRO seriously affects the nutritional value of *Brassica* vegetables.

Glucosinolate consists of a β-D-thioglucosyl, a sulfonate, and a branched R group [8]. The glucosinolates are classified into three types, namely, aliphatic, aromatic, and indole GSLs, depending on the origin of the biosynthetic precursor standard amino acid [9]. The biosynthesis pathway is divided into three steps: the carbon chain elongation of the amino acid, the formation of the basic skeleton, and the modification of the side chain [10], which results in the chemical diversity of glucosinolates [11]. Multiple QTL sites controlling the chemical structure of the side chain have been identified, including *GSL-Elong*, *GSL-OX, GSL-ALK,* and *GSL-OH* in Arabidopsis. *GSL*-*Elong* is involved in the production of glucosinolate with three carbon or four carbon side chains [12]; then, the GSL-Alk enzyme catalyzes the glucoraphanin to gluconapin [13]. Finally, the enzyme regulates the conversion of gluconapin (NAP) to progoitrin by the gene located on the *GSL-OH* locus [14,15] (Figure 1A).

The *GSL-OH* locus was finely mapped using several populations of the *A. thaliana* recombinant inbred line Ler×Cvi and a gene named *2-ODD* (2-oxo acid -dependent dioxygenase, At2g25450) was identified in Arabidopsis [16]. The expression of *2-ODD* in *A. thaliana* is positively correlated with the accumulation of PRO in different genotypes and knocking out the *GSL-OH* allele inhibits PRO synthesis in *Arabidopsis* [16]. This study provides a basis for the mechanism involved in regulating the content of PRO glucosinolates through genetic engineering. However, the function of the 2-ODD gene in other cruciferous plants is still unknown.

Chinese kale (*Brassica olearcea* var. *chinese* Lei), which originated in South China and spread throughout Southeast Asia thereafter, is an important cruciferous vegetable crop in China, especially in South China, where the majority of varieties are grown [17]. Chinese kale is rich in nutritional compounds, including vitamin C, minerals, and glucosinolates [18,19], with a long history of cultivation and abundant germplasm resources in China. Notably, the content of glucoraphanin, whose isothiocyanate product can protect against tumorigenesis, is accumulated at a high level in many genotypes of Chinese kale [20].

In our previous study, glucosinolate was extracted and analyzed in more than 70 Chinese kale genotypes, and the data showed that the ratio of PRO/NAP was significantly different in various genotypes [21] (Figure 1B). However, the mechanism of the GSL accumulation with different levels remains unknown. In this study, the PRO and NAP content of several Chinese kale genotypes was analyzed. Four ODD genes were cloned and analyzed from Chinese kale plants. The function of *BocODD1* and *BocODD2* involved in PRO biosynthesis was analyzed by overexpression and RNA interference technology. The results provide a theoretical reference for the creation of low-PRO germplasm resources by genetic engineering in Chinese kale and other *Brassica* plants.

## 2. Results

### 2.1. Determination of Glucosinolate Content in Chinese Kale

Four accessions of Chinese kale were selected from our previous study according to the glucosinolate content, in which the genotypes Shengzhong and Xichangzhong have a much higher ratio of PRO/NAP than Taoshanzhonghua and Benditiezhong (Figure 1). To further verify these results, the content of PRO was determined again, and there were 0.15, 0.16, 0.16, and 0.22 mg·g^−1^ DW PRO and 11.92, 1.43, 7.01, and 4.00 mg·g^−1^ DW NAP in the genotypes Taoshanzhonghua, Shengzhong, Benditiezhong, and Xichangzhong, respectively (Figure 2A,B). The results showed that the highest ratio of PRO/NAP was in Shengzhong, and the lowest one was in Taoshanzhonghua (Figure 2C). Although the accumulation of PRO has no significant difference in the three varieties, NAP was the highest in Taoshanzhonghua (Figure 2A), which resulted in the lowest ratio of PRO/NAP in this genotype.

To analyze the contents of PRO and NAP in the roots, stems, leaves, and flowers from the Chinese kale of the Shengzhong variety, HPLC analysis was used to investigate the glucosinolate components. In this study, nine glucosinolates were clearly detected (Appendix A), including 2-hydroxybut-3-enyl-GSLs (progoitrin, PRO), 4-(methylsulfinyl)butyl GSLs (glucorasphanin, RAA), 2-propenyl GSLs (sinigrin, SIN), but-3-enyl GSLs (gluconapin, NAP), 4-hydroxyindol-3-ylmethyl GSLs (4-hydroxyglucobrassicin, 4OH), 4-pentenyl GSLs (glucobrassicanpin, GBN), indol-3-ylmethyl GSLs (glucobrassicin, GBC), 4-methoxyindol-3-ylmethyl GSLs (4-methoxyglucobrassicin, 4ME), and 1-methoxyindol-3-ylmethyl GSLs (neoglucobrassicin, NEO) [22,23]. The content of PRO was 0.36, 0.23, 0.87, and 0.99 mg·g^−1^ DW in the leaves, stems, roots, and flowers, respectively (Figure 2D). The content of NAP was 1.63, 5.19, 11.29, and 15.85 mg·g^−1^ DW in these four tissues (Figure 2E). The ratio of PRO/NAP was the highest in the leaves, followed by the roots (Figure 2G). The total glucosinolate content was the highest in the roots, and then the flowers, stems, and leaves (Figure 2F).

The content of PRO and NAP was investigated in Chinese kale seedlings treated by high and low temperatures for three days. The results showed that the PRO content increased significantly in both the leaves and roots treated by a high or low temperature (Figure 2J,K). The level of NAP, the precursor substance of PRO, also increased significantly in the leaves and roots after 40 °C treatments (Figure 2I). The NAP content increased significantly in the leaves after 4 °C treatments but did not change significantly in the roots after 4 °C treatments (Figure 2H).

### 2.2. Isolation of ODD Homologues and the Promoters from Chinese Kale

The *Brassica oleracea* ODD genes (LOC106342562, LOC106342186, and LOC106342567) were used as reference sequences for primer design. Four cDNA sequences were cloned from the genotypes Shengzhong and Taoshanzhonghua (designated as *BocODD1*, *BocODD2, BocODD3,* and *BocODD4*). The sequences of four cDNA were compared, but SNPs were not identified between these two genotypes. Meanwhile, the corresponding DNA sequences and promoter sequences were obtained by electronic cloning (Appendix A). The open-reading fragment (ORF) varied from 1077 to 1086 bp (Figure 3B), encoding proteins of 359–361 amino acids with a predicted molecular mass of 30122.56–40230.23 kDa (Appendix A). The instability index of the four genes was no more than 40, which indicated they were stable proteins. The values of the grand average of hydropathicity (GRAVY) were negative (Appendix A), showing that they were all hydrophilic proteins.

The promoter sequences of four *BocODDs* were analyzed on the PlantCARE website. Besides some basic response elements (light-response element, TATA-box, endosperm expression, etc.), the promoter sequences also contain hormone response elements, for example, the cis-acting regulatory element involved in MeJA-responsiveness, auxin responsiveness, abscisic acid responsiveness, and a gibberellin-responsive element. Importantly, a series of MYB and MYC element sites were identified, which may be involved in drought-inducibility, light response, or other functions (Figure 3A).

### 2.3. Amino Acid Sequence Analysis of BocODD Genes

The alignment analyses indicated that the four deduced *BocODD* proteins were highly conserved and all of them contained a conserved domain of oxoglutarate/iron-dependent dioxygenase (Figure 4). The amino acid sequences of *BocODDs* shared 75–82% similarity with the ODD protein from Arabidopsis (At2g25450) (Appendix A). The BocODDs sequences were queried for similarity against the GenBank data using the BLASTP program. The homologous sequences obtained were mainly from crucifer plants, such as *Arabis alpina*, *Brassica cretica*, *Capsella rubella*, *Brassica oleracea*, and so on. A phylogenetic tree was constructed, which indicated that BocODD1~3 shared the same clade with the proteins from the *Brassica* plants including *B. oleracea*, *B. rapa*, and *Br. napus*. Interestingly, unlike BocODD1-3, BocODD4 was clustered with amino acid sequences from *C. rubella*, *R. sativus*, and *C. sativa* (Figure 5), which indicated that the function of *BocODD4* may be different from the other three BocODDs and AtODD (AT2G25450). In addition, the amino acid sequences of BocODD3 were about 90 amino acids shorter than BocODD1 and BocODD2, which indicated that BocODD3 may not be involved in glucosinolate biosynthesis. According to the results above, we chose *BocODD1* and *BocODD2* for a further analysis of their gene functions.

### 2.4. Expression Analysis of BocODD1/2 in Chinese Kale

The expression patterns of *BocODD1* and *BocODD2* were analyzed in the Chinese kale genotypes Shengzhong, Taoshanzhonghua, Benditiezhong, and Xichangzhong. The results showed that *BocODD1* and *BocODD2* were most highly expressed in the genotype Shengzhong and had lower expression levels in the other genotypes (Figure 6A). The expression levels of *BocODD1* and *BocODD2* in the roots, stems, leaves, and flowers were detected. *BocODD2* had the highest expression levels in the leaves followed by the roots (Figure 6B). The expression level of *BocODD1* was higher in the roots and flowers than leaves and stems (Figure 6B).

Glucosinolate is usually involved in the stress response in plants. Therefore, the expression of *BocODD1* and *BocODD2* may be induced by abiotic stress. In the present study, the expression levels of *BocODD1* and *BocODD2* increased significantly after 40 °C and 4 °C treatments compared to the control plants (25 °C treated for 4 h). The expression levels of the two genes increased about 3–10 times after treatment at 4 °C and 40 °C compared to the control plants (Figure 6C).

### 2.5. Function Analysis of BocODD1/2 in the Conversion of 3-Butenyl Glucosinolate to 2-Hydroxy-3-Butenyl Glucosinolate

To study the function of the *BocODD1* and *BocODD2* genes in Chinese kale, over-expression and RNAi were performed on the plants using an *Agrobacterium tumefaciens*-mediated transformation. Two *BocODD1* overexpression lines (OE-ODD1-1 and OE-ODD1-8), two *BocODD2* overexpression lines (OE-ODD2-3 and OE-ODD2-5), and two RNAi lines (RNAi-ODD-9 and RNAi-ODD-26) were used for further study. Their T1 plants were used for phenotypic and molecular analyses. The morphologies of the transgenic plants and wild type (WT) plants had no significant difference with respect to leaf type, plant type, plant height, color, etc. (Figure 7D–F).

The expression level of *BocODD1* in the transgenic plants OE-ODD1-1 and OE-ODD1-8 were higher than the wild type plants. The expression level of *BocODD2* was 2.4 and 2.5 times higher, respectively, in the OE-ODD2-3 and OE-ODD2-5 compared to the wild type plants (Figure 7G,H). The expression level of *BocODD1* and *BocODD2* in RNAi-ODD-9 was one-sixth that of the WT, while the expression level of *BocODD1* was significantly lower in RNAi-ODD-26 compared to that of the WT plants. The expression level of *BocODD2* in both of the RNAi plants was significantly lower compared to the WT plants (Figure 7I).

To identify the function of *BocODD1*, *2* in the biosynthesis of glucosinolate, the concentrations of PRO and NAPwere determined in transgenic plants. The results showed that the content of PRO was 0.62 and 1.17 mg·g^−1^ DW and the ratios of PRO/NAP were 0.07 and 0.13 in the OE-ODD1-1-2 and OE-ODD1-8-11 plants, respectively, which were significantly higher than the WT1 plants (0.03 ). However, the content of NAP was not significantly changed in the overexpression plants OE-ODD1-1-2 and OE-ODD2-3-11 (8.73 and 8.77 mg·g^−1^ DW) compared to the WT1 plants (7.44 mg·g^−1^ DW). The level of PRO was significantly decreased in RNAi-ODD-9 (0.04 mg·g^−1^ DW) and RNAi-ODD-26 (0.04 mg·g^−1^ DW) compared to the WT2 plants (0.08mg·g^−1^ DW). The NAP was significantly higher in RNAi-ODD-9 (2.47 mg·g^−1^ DW) and RNAi-ODD-26 (3.36 mg·g^−1^ DW) compared to the WT2 plants (0.99mg·g^−1^ DW), which also resulted in the PRO/NAP level being significantly lower (Table 1).

## 3. Discussion

Glucosinolates are a large group of plant secondary metabolites with nutritional effects and are mainly found in cruciferous plants [24]. As a main source of bitterness in cruciferous vegetables, the degradation product of 2-hydroxy-3-butenyl glucosinolate has biological activities including toxicity towards *Caenorhabditis elegans*, the inhibition of seed germination, the induction of goiter disease in mammals, and the production of bitter flavors in *Brassica* vegetable crops [6,16,25], which limit the consumption of cruciferous vegetables.

It has been reported in Arabidopsis that 2-hydroxy-3-butenyl glucosinolate is mainly converted from 3-butenyl glucosinolate, which is ubiquitous in cruciferous plants [16,24]. The fine-scale mapping of the *GSL-OH* locus identified a 2-oxoacid-dependent dioxygenase (ODD) encoded by At2g25450 required for the formation of 2-hydroxybut-3-enyl glucosinolate [16,24]. In the present study, we cloned four *BocODD* genes from Chinese kale. *ODD* encodes 2-oxoacid-dependent dioxygenase, which belongs to the 2-oxoglutarate-dependent dioxygenase superfamily that participates in secondary metabolites’ biosynthesis, transport, and catabolism [26]. The deduced protein sequences of the four *BocODDs* contained Fe(^2+^) and the 2-oxoglutarate (2OG)-dependent dioxygenase domain, which catalyze the oxidation of an organic substrate using a dioxygen molecule, mostly by using ferrous iron as the active site cofactor and 2OG as a co-substrate [27]. The 2OG gene family can be classified into three classes—DOXA, DOXB, and DOXC—based on their amino acid sequences [28]. The DOXC class is involved in the biosynthesis of various phytochemicals [28]. ODDs have been classified into DOXC and catalyze the biosynthesis of goitrogenic 2-hydroxy-3-butenyl glucosinolate in Arabidopsis [16,28]. However, there was no difference in the DNA and cDNA sequences between the two Chinese kale genotypes, indicating that the differences in the 2-hydroxy-3-butenyl glucosinolate content were not caused by gene sequences.

A series of hormone response elements were identified in the promoter sequences, which indicated that *BocODD* genes might be involved in hormone response and stress regulation. It was reported that *MYB28/MYB29/MYB76* were involved in the regulation of aliphatic glucosinolates synthesis [29], and can positively regulate the expression of glucosinolates’ biosynthesis genes. In the present study, the MYB and MYC sites identified in the promoter sequences suggested that *BocODD* genes may be regulated by MYB and MYC transcription factors, which may result in the variable content of 2-hydroxy-3-butenyl glucosinolate in Chinese kale.

By analyzing glucosinolate content and gene expression patterns, we found that the ratio of PRO/NAP in Shengzhong was the highest and lowest in Taoshanzhonghua, which was consistent with the results of our previous study [21], and which was also consistent with the trend of the *BocODD1*, *2* expression patterns. There was no significant difference in the content of PRO glucosinolate between the two genotypes, Xichangzhong and Benditiezhong, but the NAP content was much higher in the genotype Xichangzhong than Benditiezhong, which resulted in the significant difference in the PRO/NAP ratio between the genotype Xichangzhong and Benditiezhong. The content of glucosinolates in the tubers and leaves was analyzed in the *B. rapa* turnip, and the content of PRO in the tubers was much higher than that in the leaves [30]. In the present study, the trend of *BocODD1* expression patterns was almost the same as the PRO/NAP ratio in the different tissues and the content of PRO was also higher in the roots than in the leaves. The roots had the highest glucosinolate content in all the examined tissues, which is consistent with the results reported previously [31]. More glucosinolates continue to resist the pathogenic microorganisms in the soil and some insects, and most of the glucosinolate synthesis genes showed higher expression levels in the roots than other tissues [32]. Meanwhile, ODD genes play an important role in root formation [33], which is up-regulated during adventitious root formation in apples [34].

Cold and heat stress can enhance the accumulation of aliphatic glucosinolates [35]. In the present study, the expression level of *BocODD1/2* was increased significantly in the roots and leaves after high and low temperature treatments. However, the ratio of PRO/NAP was not significantly changed in the leaves after cold or heat stress. We further identified the content of glucosinolate in the roots, and the results showed that the content of PRO and NAP glucosinolate was significantly increased, as was the ratio of PRO/NAP, which indicated that the PRO in roots may play an important role in the cold resistance and heat resistance in Chinese kale.

The gene at the *GSL-OH* site has been identified in Arabidopsis as being involved in the synthesis of PRO glucosinolate [16]. In the present study, the overexpression of *BocODD1* or *BocODD2* resulted in a significant increase in PRO content and the ratio of PRO/NAP. Meanwhile, the knockdown of *BocODD1* or *BocODD2* resulted in the significant decrease in PRO content and the ratio of PRO/NAP, which suggested that *BocODD1* and *BocODD2* were involved in the conversion of 3-butenyl glucosinolate to 2-hydroxy-3-butenyl glucosinolate. In summary, *BocODD1* and *BocODD2* may play important roles in the biosynthesis of PRO glucosinolates in Chinese kale.

## 4. Materials and Methods

### 4.1. Gene Cloning and Sequence Analysis

The sequences of ODD genes in Chinese kale were obtained in *B. oleracea* from *Brassica* Database (http://brassicadb.org/brad/(accessed on 30 Augest 2016) and in transcriptome data of the Chinese kale [32]. Gene cloning was performed from total genomic DNA of Chinese kale genotypes “Shengzhong” and “Taoshanzhonghua” using gene-specific primers (Appendix A). Polymerase Chain Reaction was performed with 32 cycles (94 °C for 30 s, 55 °C for 30 s, and 72 °C for 2 min) and a final extension at 72 °C for 5 min. Polymerase Chain Reaction products were cloned into a pMD19-T vector, and then transformed into *Escherichia coli* DH5α competent cells. Plasmids were isolated and genomic sequence was validated. The phylogenetic tree was constructed using the neighbor-joining method with MEGA (version 6.0) software [36], and was modified in iTOL (https://itol.embl.de/help.cgi#boot)(accessed on 20 September 2019).

### 4.2. Expression Analysis

Chinese kale seeds of genotypes Shengzhong, Taoshanzhonghua, Xichangzhong, and Benditiezhong were cultured in a greenhouse at 22–25 °C in South China Agricultural University (Guangzhou, China). Two fully expanded young leaves with no pests and no wounds were sampled at the six-leaf stage. The expression patterns of *BocODD1* and *BocODD2* and the content of GSL were analyzed. Four tissues (roots, stems, middle leaves, and flowers) of genotype shengzhong at flowering stage under normal growth conditions were collected for qRT-PCR and glucosinolate analysis. For heat and cold stress, the Chinese kale Shengzhong seeds were briefly surface-sterilized with 2% sodium hypochlorite and sowed in 1/2 Murashige and Skoog medium (MS, pH 5.8) [37], and then placed in a greenhouse. Six days later, the cotyledons were fully expanded. Ten bottles of seedlings with the same growth conditions were treated at 4 °C, 25 °C, and 40 °C. The roots and leaves of seedlings treated after 4 h were collected for the expression patterns analysis. The roots and leaves of seedlings treated after 3 days were collected for glucosinolate content determination.

Total RNA was isolated using a Total RNA Extraction Kit according to the manufacturer′s instructions. First-strand cDNAs were synthesized using TransScript First-Strand cDNA Synthesis SuperMix with oligo(dT) as a primer in a 20 µL reaction. Quantitative real-time RT-PCR was carried out using specific primers for each gene (Appendix A). The reaction was performed in LightCycle 480 (Roche) using SYBR green protocol. *Actin2* was used as an internal reference gene. Amplification was carried out with the following cycling parameters: denaturing for 2 min at 95 °C, 40 cycles of denaturation at 95 °C for 15 s, annealing for 30 s at 58 °C, and extension at 72 °C for 30 s. The same experiments were repeated in triplicate for each sample, and the relative expression was calculated by the 2^−△△Ct^ method.

### 4.3. Glucosinolate Extraction and Analysis

The extraction and analysis of glucosinolates were based on previous study [21,38] and the ISO 9167-1 method [39]. Briefly, 0.20 g of freeze-dried samples was extracted twice with 70% methanol, employing 100 μL benzyl glucosinolate as the internal standard. The extract was filtered and desulphated overnight at room temperature on DEAE Sephadex A-25 columns using purified sulfate. Then, the obtained glucosinolate solution was filtered through a 0.22 μm membrane. A total of 20 μL of glucosinolate was analyzed by HPLC-UV (waters 2695) equipped with C18 column (4.6 × 250 mm, 5 μm, and waters) at 229 nm. The mobile phases A (ultrapure water) and B (acetonitrile) had varied gradients: 0–32 min and 0–20% acetonitrile; 32–38 min and 20% acetonitrile; 38–40 min and 20–100% acetonitrile. Flow rate was 1.0 mL/min and column temperature was 30 °C. The obtained HPLC diagram is shown in Appendix A. Mass spectrometry analysis (HPLC/MS, Agilent 1100 series, Agilent Technologies) was used to identify individual GSLs. MS analysis settings were as follows: positive ion mode (scan range m/z 100–600), nebulizer pressure was set at 60 psi, gas-drying temperature at 350 °C, and capillary voltage at 4 kV. The glucosinolate content was calculated according to the formula: D = (Ai/As) × F × Ns/M (D, the content of glucosinolate; Ai, peak area of glucosinolate component; As, peak area of internal standard; F, the relative response factor of desulfoglucosinolate relative to the internal standard [40]; Ns, amount of internal standard; M, dry weight of plant materials). Three biological replicates and three technical replicates were conducted for each sample. The unit of internal standard concentration is mg/mL, and the unit of glucosinolate content calculated in this experiment is mg/g dry weight (DW).

### 4.4. Transformation of BocODD1 and BocODD2 in Chinese Kale

The coding sequences of *BocODD1* and *BocODD2* were isolated and amplified using Shengzhong genotype’s genomic cDNA as template with gene-specific primers including restriction sites (BamHI/ScaI). The amplified product was inserted into a pBI121 vector driven by the cauliflower mosaic virus (CaMV) 35S promoter. A 338-bp sequence from the common sequence of *BocODD1* and *BocODD2* was cloned into the pFGC5941 vector for RNAi analysis. The primers are listed in Appendix A. The recombinant vector was verified by DNA sequencing and subsequently transformed into EHA105. Transformation of the Chinese Kale lines (Laozhongxianggu) was carried out following the method of our previous study [37] with minor modifications. Chinese kale seeds were surface-sterilized with 70% ethanol for 90 s following 2% sodium hypochlorite for 9 min and then rinsed three times in sterile, distilled water. Sterilized seeds were placed in 1/2 MS medium for 5 days. Hypocotyl explants were cultured for 2 days in MS medium containing 2 mg/L 6-benzylaminopurine (6-BA) and 0.05 mg/L naphthaleneacetic acid (NAA). The precultured hypocotyls were immersed in an overnight-cultured suspension of *Agrobacterium tumefaciens* strain EHA105 for 8 min. After 72 h in co-cultivation, the cotyledons were transferred to selection medium supplemented with 300 mg/L Cef,2 mg/L 6-BA, 0.05 mg/L NAA,12 mg/L kanamycin (Ka), or 10 mg/L hygromycin (Hyt).

After 4–5 weeks, the seedlings with vigorous roots were transferred to the greenhouse for cultivation. The matured seeds were collected from a single plant and sown in 50-hole plug trays produce T1 plants. The tender and fully expanded leaves of T1 generation plants were sampled for DNA and RNA extraction. Polymerase Chain Reaction (PCR) was conducted to identify transgenic plants. Quantitative real-time PCR (qRT-PCR) and HPLC analysis were performed to determine the specific gene expression and glucosinolate content, respectively.

## 5. Conclusions

Our results suggest that there are four ODD genes in Chinese kale, in which *BocODD1* and *BocODD2* are the most important candidate genes involved in glocosinolate biosynthesis according to the results of the sequence and expression analyses and the glucosinolate content determination. Further study via a gene function analysis showed that both *BocODD1* and *BocODD2* regulate the biosynthesis of progoitrin in Chinese kale. The study of BocODD genes not only helps to understand the molecular mechanism of glucosinoalte biosynthesis in Chinese kale but also provides a theoretical reference for the creation of low-PRO germplasm resources through genetic engineering in Chinese kale and other *Brassica* plants.

## Figures and Tables

**Figure 1 ijms-23-14781-f001:**
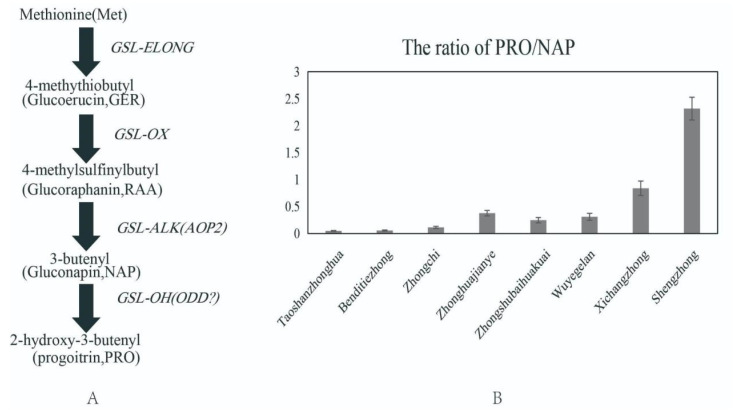
The conversion of NAP to PRO glucosinolate. (**A**) The side chain of aliphatic glucosinolate biosynthesis; (**B**) the ratio of PRO/NAP in different Chinese kale genotypes.

**Figure 2 ijms-23-14781-f002:**
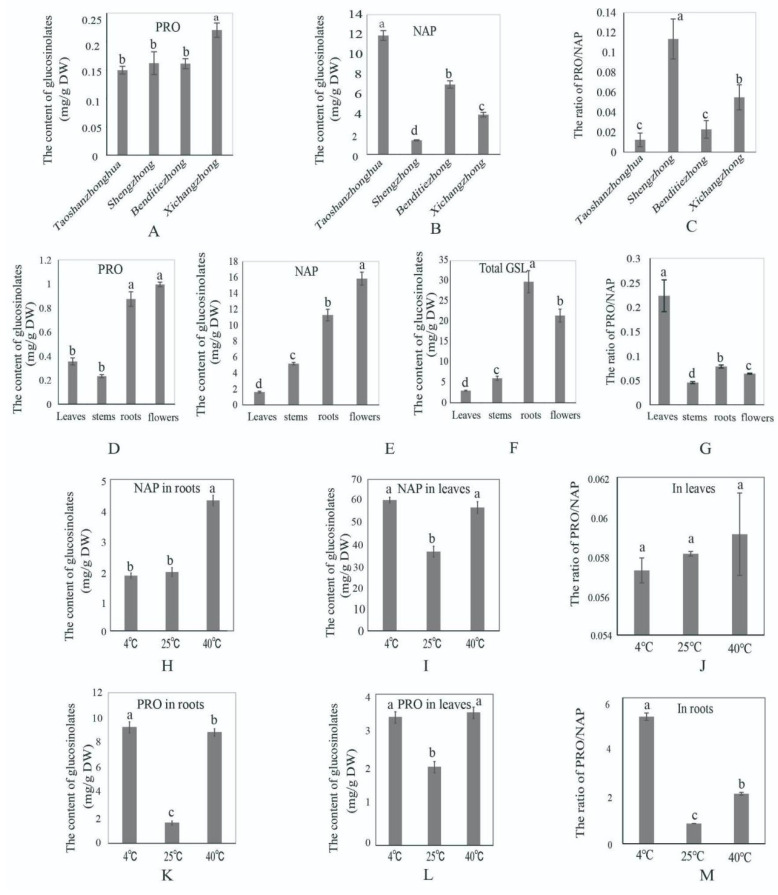
The determination of glucosinolates. (**A**–**C**) the content of PRO and NAP glucosinolates and the ratio of PRO/NAP in four different genotypes; (**D**–**G**) the content of PRO glucosinolate, NAP glucosinolate, total glucosinolates, and the ratio of PRO/NAP in four tissues; (**H**,**I**) the content of NAP glucosinolate in roots and in leaves after cold and heat stress; (**J**) the ratio of PRO/NAP in leaves after cold and heat stress; (**K**,**L**) the content of PRO glucosinolates in roots and in leaves after cold and heat stress; (**M**) the ratio of PRO/NAP in roots after cold and heat stress. Different letters (a, b, c, d) indicate statistically significant difference between samples (one-way ANOVA, Tukey’s test *p* < 0.05).

**Figure 3 ijms-23-14781-f003:**
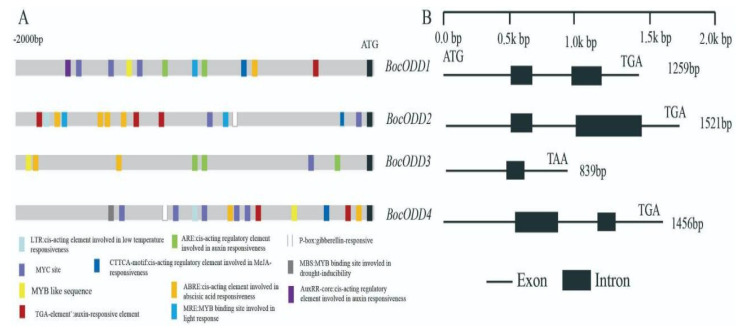
The structure of four ODD genes and their promoters in Chinese kale. (**A**) The structure of promoters; (**B**) the structure of DNA sequence.

**Figure 4 ijms-23-14781-f004:**
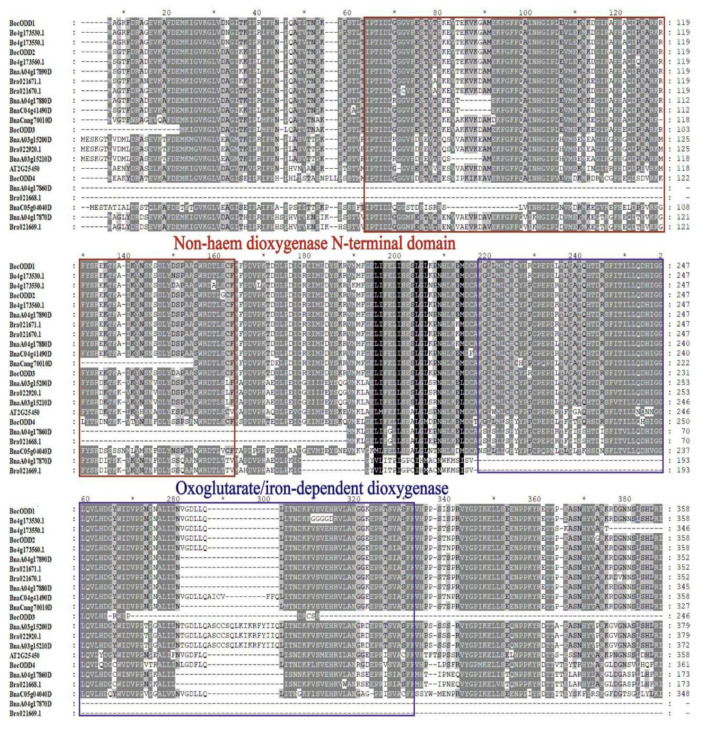
Amino acid sequence alignment of ODD proteins from Chinese kale, *Brassica oleracea*, *Brassica rapa*, *Brassica napus,* and *Arabidopsis thaliana*. The non-haem dioxygenase N-terminal domain and oxoglutarate/iron-dependent dioxygenase are boxed in red and blue, respectively.

**Figure 5 ijms-23-14781-f005:**
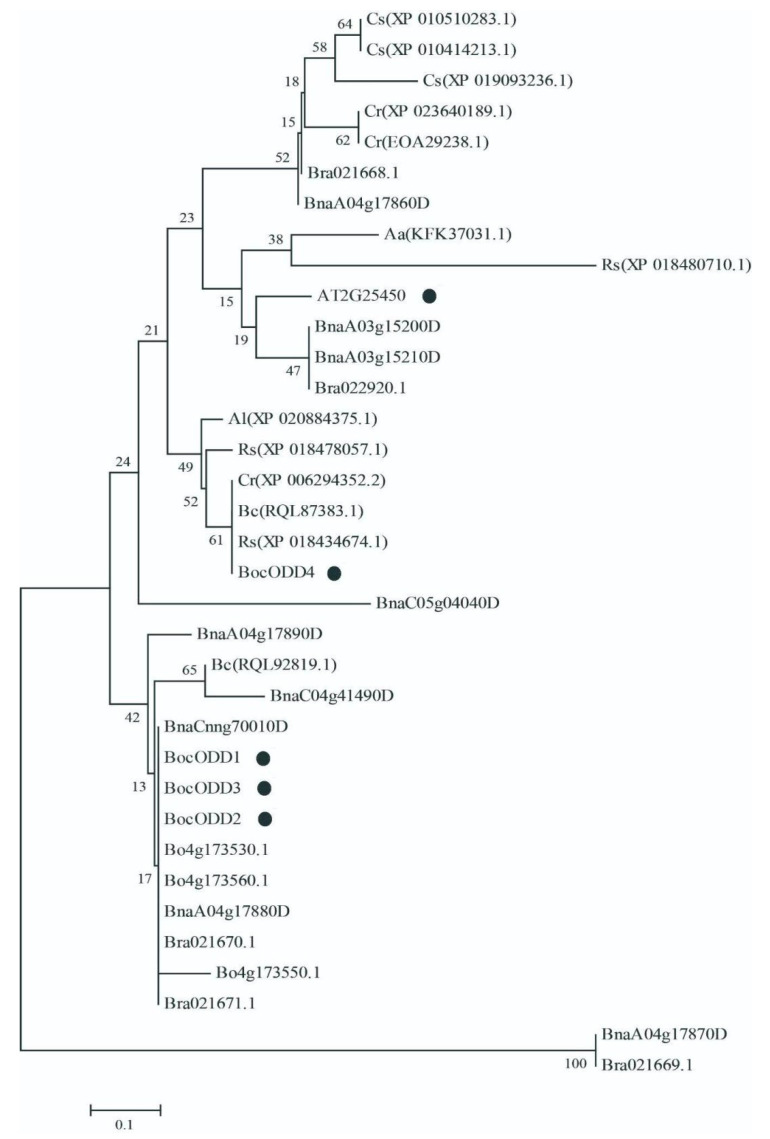
Phylogenetic analysis of ODDs in the *Brassica* family. This tree was constructed by MEGA 6.06 software. Bootstrap values with 1000 replicates are given as percentages; ODD genes were listed in Appendix A.

**Figure 6 ijms-23-14781-f006:**
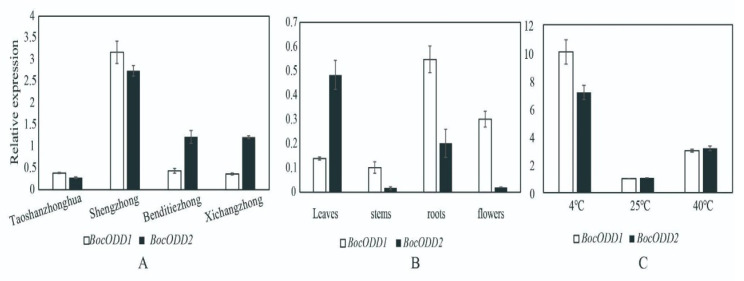
Expression analysis of *BocODD1/2* in Chinese kale. (**A**) the expression levels of *BocODD1* and *BocODD2* in four genotypes—Shengzhong, Taoshanzhonghua, Benditiezhong, and Xichangzhong; (**B**) the expression levels of *BocODD1* and *BocODD2* in leaves, stems, roots, and flowers; (**C**) the expression levels of *BocODD1* and *BocODD2* in leaves treated for 4 h at 4 °C, 25 °C, and 40 °C.

**Figure 7 ijms-23-14781-f007:**
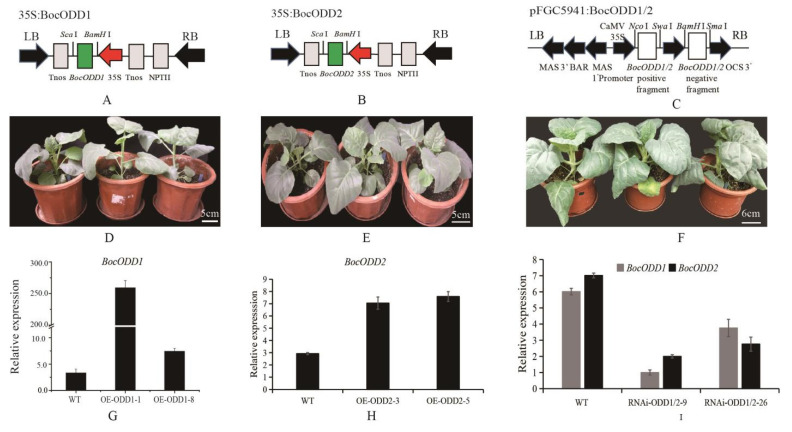
Functional analysis of *BocODD1/2*. (**A**) The structure of overexpression vector for *BocODD1*; (**B**) the structure of overexpression vector for *BocODD2*; (**C**) the structure of RNAi vector for *BocODD1/2*; (**D**–**F**) the phenotypes of overexpression and RNAi transgene plants; (**G**) and (**H**) the expression levels of *BocODD1* or *BocODD2* in overexpression plants, respectively; (**I**) the expression levels of *BocODD1* or *BocODD2* in RNAi plants.

**Table 1 ijms-23-14781-t001:** The content of glucosinolates of *BocODD1/2* transformed plants.

Strain	Generation	PRO (mg/g DW)	NAP (mg/g DW)	The Ratio of PRO/NAP
WT1		0.20	7.44	0.03
OE-ODD1-1-2	T1	0.62 **	8.73	0.07 **
OE-ODD1-8-11	T1	1.17 **	8.77	0.13 **
OE-ODD2-3-11	T1	1.40 **	6.47	0.19 **
OE-ODD2-5-8	T1	1.18 **	6.34	0.22 **
WT2		0.08	0.99	0.08
RNAi-9	T0	0.04 **	2.47 **	0.02 **
RNAi-26	T0	0.04 **	3.36 **	0.01 **

^**^, Student’s *t*-test significant at *p* < 0.01

## Data Availability

Not applicable.

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
