# Peer review of "BocODD1 and BocODD2 Regulate the Biosynthesis of Progoitrin Glucosinolate in Chinese Kale"

_ijms, 2022, doi:10.3390/ijms232314781_

Round 1

Reviewer 1 Report

Dear Author

Regarding to the manuscript ID: ijms-2018874

Title: BocODD1 and BocODD2 regulate the biosynthesis of Progoitrin glucosinolate in Chinese kale

- Please have a look through out the manuscript and concern to all the comments

- The references need to  corrected according to the journal format 

Best regards

Author Response

Dear reviewer:

We want to thank you for the comments and critiques. Accordingly, we have made many changes in the revised manuscript according to your suggestion. Please see attached documents of revised manuscript.

The references format will be revised by the journal editors.

Reviewer 2 Report

The problem in the paper is very interesting. The obtained results of the tested genes are included in the synthesis of PRO, which in the root plays an important role in resistance to abiotic stress. GLS is important for resisting soil pathogenic microorganisms and insects. Therefore, further research should be continued.

Author Response

Dear reviewer: We thank you for the comments. Accordingly, we have made many changes in the revised manuscript. Please see attached documents of revised manuscript.

By the way, the references format will be revised by the journal editors.

Reviewer 3 Report

BocODD1 and BocODD2 Regulate the Biosynthesis of Progoitrin Glucosinolate in Chinese Kale

The authors presented the function of BocODD1 and BocODD2 in biosynthesis of glucosinolates. The content of progoitrin and gluconapin in several Chinese kale genotypes were analyzed. Here are the major concerns:

1.      The authors reported presence of 9 glucosinolates in Chinese kale.

a) How these glucosinolates were identified? This should be also included in Materials and methods. Also this statement in the manuscript is connected to Figure S1, where is that figure?

b) Names of glucosinolates should be corrected according to IUPAC. Literature for these compounds is not up to date (see Blažević et al., Glucosinolate structural diversity, identification, chemical synthesis and metabolism in plants, Phytochemistry, Volume 169, 2020).

c) Materials and methods: HPLC analysis of glucosinolates; The authors used wrong reference for method description. Please, report the response factors used for glucosinolate quantification.

2.      Many mistakes throughout the manuscript:

a)  check all the words that should be italic

b)  abbreviation for glucosinolates should be the same, for example in: “In this study, 9 glucosinolates were clearly detected (Figure S1), including 2-hydroxy-3-butenyl-GS (progoitrin, PRO), 4-methylsulfinylbutyl-GS” - Why suddenly using GS abbreviation for glucosinolate instead of GSL? (line numbers are missing for easier reading and commenting).

Author Response

Dear reviewer: Thank you for the comments and critiques. Accordingly, we have made many changes in the manuscript. Please find below the specific response to each question.

  1. The authors reported presence of 9 glucosinolates in Chinese kale.
  2. a)How these glucosinolates were identified? This should be also included in Materials and methods. Also this statement in the manuscript is connected to Figure S1, where is that figure?

Answer:We thank the reviewer for this comment. indeed, we have ignored this part of the content, and now we have added the method for glucosinolates detection to the methods and materials part. The figure S1 showed the HPLC diagram of glucosinolate detection. 

  1. b) Names of glucosinolates should be corrected according to IUPAC. Literature for these compounds is not up to date (see Blažević et al., Glucosinolate structural diversity, identification, chemical synthesis and metabolism in plants, Phytochemistry, Volume 169, 2020).

Answer:Thank you for this comment. The names of glucosinolates have been corrected according to IUPAC in the revised manuscript.

  1. c)Materials and methods: HPLC analysis of glucosinolates; The authors used wrong reference for method description. Please, report the response factors used for glucosinolate quantification.

Answer:Thank you for your good comments. We used the wrong reference for method description because the wrong insertion of literature number. Now we have revised the methods for extraction, detection and identification of glucosinolate and corrected the related literature. The response factors used for glucosinolate quantification have also been added in the revised manuscript.

  1. Many mistakes throughout the manuscript:
  2. a)check all the words that should be italic
  3. b)abbreviation for glucosinolates should be the same, for example in: “In this study, 9 glucosinolates were clearly detected (Figure S1), including 2-hydroxy-3-butenyl-GS (progoitrin, PRO), 4-methylsulfinylbutyl-GS” - Why suddenly using GS abbreviation for glucosinolate instead of GSL? (line numbers are missing for easier reading and commenting).

Answer:Thank you for your good comments. We have checked and corrected all the words should be italic throughout the revised manuscript. “GSL” was used for abbreviation for glucosinolate throughout the revised manuscript.

Round 2

Reviewer 2 Report

All suggestions are accepted. Therefore, he considers that the paper has been solidly done and I suggest that it be accepted in its current form.

Reviewer 3 Report

Materials and methods: 4.3. Glucosinolate extraction and HPLC analysis):

There are still missing some important information regarding method for the glucosinolate quantification, for example, HPLC instrument used.  Please provide detailed description, meaning specify values of the response factors used.

Figure S1. HPLC chromatogram of glucosinolate – this should be HPLC chromatogram of desulfoglucosinolates?

“2-hydroxybut-3-enyl glucosinolates; 2. 4-(methylsulfinyl)butyl glucosinolates; 3. 2-propenyl glucosinolates; 4. but-3-enyl glucosinolates” – “glucosinolates” is plural… Please correct: 4-(methylsulfinyl)butyl glucosinolate etc.

Author Response

Comments and Suggestions for Authors

  1. Materials and methods: 4.3. Glucosinolate extraction and HPLC analysis):

There are still missing some important information regarding method for the glucosinolate quantification, for example, HPLC instrument used.  Please provide detailed description, meaning specify values of the response factors used.

Answer: Thank you for this comment.  We tried our best to understand your question and added the method for the glucosinolate quantification to the Materials and methods part as follows.

The extraction and analysis of glucosinolates were based on previous study [21, 38] and the ISO 9167-1 method [39]. Briefly, 0.20 g of freeze-dried samples were extracted twice with 70% methanol, with 100 μl benzyl glucosinolate as the internal standard. The extract was filtered and desulphated overnight at room temperature on DEAE Sephadex A-25 columns using purified sulfate. Then, the obtained glucosinolate solution was filtered through a 0.22 μm membrane. Twenty μl of glucosinolate is analyzed by HPLC-UV (waters 2695) equipped with C18 column (4.6×250mm, 5μm, waters) at 229nm. The mobile phases A (ultrapure water) and B (acetonitrile) change in gradient: 0-32 min, 0-20% acetonitrile; 32-38 min, 20% acetonitrile; 38-40 min, 20-100% acetonitrile. Flow rate is 1.0 ml/min and column temperature is 30℃. The obtained HPLC diagram was shown as Figure S1. Mass spectrometry analysis (HPLC/MS, Agilent 1100 series, Agilent Technologies) was used to identify individual GSLs. MS analysis settings were as follows: positive ion mode (scan range m/z 100–600). Nebulizer pressure was set at 60 psi, gas-drying temperature at 350℃, and capillary voltage at 4kV. The glucosinolate content were calculated according to the formula: D=(Ai/As)×F×Ns/M (D, the content of glucosinolate; Ai, peak area of glucosinolate component; As, peak area of internal standard; F, the relative response factor of desulfoglucosinolate relative to the internal standard [40]; Ns, amount of internal standard; M, dry weight of plant materials). Three biological replicates and three technical replicates were conducted for each sample. The unit of internal standard concentration is mg/ml, and the unit of glucosinolate content calculated in this experiment is mg/g dry weight (DW).

  1. Figure S1. HPLC chromatogram of glucosinolate – this should be HPLC chromatogram of desulfoglucosinolates?

“2-hydroxybut-3-enyl glucosinolates; 2. 4-(methylsulfinyl)butyl glucosinolates; 3. 2-propenyl glucosinolates; 4. but-3-enyl glucosinolates” – “glucosinolates” is plural… Please correct: 4-(methylsulfinyl)butyl glucosinolate etc.

Answer: Thank you for your good comments. It should be HPLC chromatogram of desulfoglucosinolates in Figure S1,and we have revised the Figure S1 in the supplementary Figures. These “4-(methylsulfinyl)butyl glucosinolates” were also corrected in the supplementary Figures.

Round 3

Reviewer 3 Report

The authors have corrected the manuscript according to comments. The paper can be recommended for publication.